# “If You Can’t Control the Wind, Adjust Your Sail”: Tips for Post-Pandemic Benefit Finding from Young Adults Living with Multiple Sclerosis. A Qualitative Study

**DOI:** 10.3390/ijerph18084156

**Published:** 2021-04-14

**Authors:** Silvia Poli, Michela Rimondini, Alberto Gajofatto, Maria Angela Mazzi, Isolde Martina Busch, Francesca Gobbin, Federico Schena, Lidia Del Piccolo, Valeria Donisi

**Affiliations:** Department of Neurosciences, Biomedicine and Movement Sciences, University of Verona, 37134 Verona, Italy; silvia.poli@univr.it (S.P.); alberto.gajofatto@univr.it (A.G.); mariangela.mazzi@univr.it (M.A.M.); isoldemartina.busch@univr.it (I.M.B.); francesca.gobbin@univr.it (F.G.); federico.schena@univr.it (F.S.); lidia.delpiccolo@univr.it (L.D.P.); valeria.donisi@univr.it (V.D.)

**Keywords:** benefit finding, resilience, pandemic, COVID-19, multiple sclerosis, youth, qualitative approach, psychology, vulnerability

## Abstract

The COVID-19 outbreak has impacted the wellbeing of people worldwide, potentially increasing maladaptive psychological responses of vulnerable populations. Although young adults with multiple sclerosis (yawMS) might be at greater risk of developing psychological distress linked to the pandemic, they might also be able to adapt to stress and find meaning in adverse life events. The aim of the present study was to explore benefit finding in response to the pandemic in a sample of yawMS. As part of a larger project, data were collected using a cross-sectional, web-based survey. Benefit finding was analysed using a qualitative thematic approach; descriptive and inferential statistics were performed to describe the sample and compare sub-groups. Out of 247 respondents with mostly relapsing-remitting MS, 199 (31.9 ± 6.97 years) reported at least one benefit. Qualitative analysis showed that during the pandemic yawMS found benefits related to three themes: personal growth, relational growth, and existential growth. No differences in benefit finding were found between age sub-groups (18–30 vs. 31–45). Participants reported a wide range of benefits, some of which seem to be specific to MS or the pandemic. Results have been transformed into tips to be introduced in clinical practice to promote resilience in yawMS through meaning making.

## 1. Introduction

The COVID-19 outbreak is a far-reaching public health crisis and unprecedented measures have been adopted across the world to contain the spread of the virus. People have to deal with direct consequences for themselves and their families (e.g., fear of getting ill, illness, grief or death) and have to adapt to conditions impacting the quality of life such as social distancing, travel restrictions or bans, closing of schools and universities, limitation of economic activities and closure of cultural and social activities [1]. Having to navigate this “Covid-19 storm” has been widely recognized as a significant psychological stressor for the general population [2,3,4,5]. However, metaphorically speaking, not all boats are the same and some people are disproportionally affected by the pandemic and particularly exposed to the risk of maladaptive psychological responses. A recent literature review highlighted that being young and having a chronic illness are some of the risk factors of distress [6].

People with chronic medical conditions can manifest a greater fear of infection and perceived stress, being more at risk of developing severe complications from COVID-19 [7]. Moreover, the pandemic has drastically affected the clinical management of patients with chronic conditions, thus potentially jeopardizing their wellbeing. Indeed, since health care systems had to rapidly adapt to acute care for COVID-19 patients and to infection control in the community, resources from chronic care were shifted away [8]. In Italy, as in many countries around the world, non-urgent care was suspended and access to in-hospital services restricted [9] with obvious effects on the continuity and quality of care [8]. This could be detrimental to people with chronic conditions characterized by uncertainty and that need regular monitoring such as multiple sclerosis (MS), a chronic neurological disease with an unpredictable clinical course, usually first diagnosed in young adults (20–40 years old) [10].

MS clinical onset typically occurs with a subacute neurological event (i.e., clinically isolated syndrome–CIS) accompanied or followed by evidence of space and time dissemination of demyelinating lesions, which can be demonstrated either clinically by relapses and disability progression or radiologically by brain and spinal cord magnetic resonance imaging [11]. Most cases exhibit clinical activity in form of relapses, which define the relapsing-remitting course of the disease; around half of these cases convert to a secondary progressive clinical course by 15 years from onset, which is characterized by irreversible accumulation of neurological disability over time independent of relapses [12]. Notably, around 15–20% of MS cases experience a gradual accrual of disability from onset with no or little evidence of relapses; this type of clinical course is known as primary progressive.

Patients with MS often consider physicians as trusted navigators who can help them to manage the uncertainty of the diagnosis and to clarify the vast amount of information patients are often exposed to [13]. Uncertainty and information overload are significant elements of the COVID-19 pandemic, especially for patients with MS that could experience concerns and doubts about their health and care, for example, regarding risks associated with immunosuppressive/modulating medication. However, with the reorganization of care and the reduction of patients-clinicians contacts, the need for trusted information could not always be met, especially during the first phase of the pandemic. Studies regarding patients with MS reported significant disruption to the management of the disease, with reorganization of care (telemedicine), canceled or postponed medical visits, and changes in pharmacological treatment [14,15].

Despite these challenges that would suggest a greater risk for developing negative emotional responses [16], studies exploring the psychological reaction of patients with MS found mixed results. Some studies highlighted high levels of anxiety [17,18] perceived stress [19] and depression [15,18,19,20,21] while others did not find any worsening of emotional distress and also an improvement in some areas of quality of life [22,23]. Indeed, although further studies with larger samples are needed, it can be hypothesized that patients with MS managed to maintain an adequate level of wellbeing. In fact, people are able to adapt to stress and even when facing highly aversive life events, resilience and growth are possible outcomes [24,25]. Consistent with this hypothesis, a study showed high levels of resilience in MS patients [26]. Resilience can be viewed as one’s ability to “bounce back” from negative life events and to adapt to stressful circumstances [27]. Post-traumatic growth is defined as the tendency to report positive transformation in the aftermath of challenging circumstances [28]. Therefore, it could be seen as “bouncing forward” [29]. Meaning-making processes are fundamental in both constructs. Benefit finding, a meaning-making construct, is defined as the identification of benefits, such as assumptions about the world and the self, in adversity and negative life events [30,31]. Studies focusing on people with MS have shown that they report a wide range of benefits associated with their illness [32] and that benefit finding has a positive effect on adjustment [33]. Research has focused on negative outcomes in the face of the current pandemic, but it is essential to investigate how individuals, in particular, vulnerable populations, such as young people with chronic disease, can maintain resilience [24]. Adopting a resource-based approach aimed at enhancing individual strength and the natural ability to adapt to stress is essential to prevent maladaptive responses, to promote wellbeing, and to foster the dynamic process of adaptation to illness, vulnerability, and chronicity.

The first aim of the present study is to explore benefit finding in response to the COVID-19 pandemic in a sample of young adults living with MS (i.e., 18–45 years old). Considering the potential differences in psychosocial adaptation to the disease and to the pandemic and in life experiences between very young adults under 30 years and those between the ages of 31 and 45, the second aim of the present research is to compare benefit finding for these two subgroups.

To gain an in-depth understanding of what young adults with MS (yawMS) have learned in the face of a pandemic and to value different nuances across individual perspectives, a qualitative approach has been applied. The results will provide insights on the resilience process put in place in the aftermath of the pandemic by this vulnerable group of patients. Moreover, results will allow to create patient-generated tips for other people with MS. These tips will be used in clinical practice in order to reinforce resilience competences and promote wellbeing during the pandemic and also regarding adjustment to illness.

## 2. Materials and Methods

The present study has a cross-sectional, observational design and follows the STROBE checklist [34]. The study is part of a bigger project “ESPRIMO”, which aims to foster resilience in yawMS [35].

### 2.1. Participants 

YawMS resident in Italy during the COVID-19 pandemic and meeting the following inclusion criteria could participate: age 18–45 years; MS diagnosis; Italian speakers; signed electronic informed consent.

### 2.2. Instruments

An ad hoc, self-administered questionnaire was developed for the purpose of the study. The questionnaire was composed of questions regarding respondents’ sociodemographic and clinical characteristics, 19 closed questions regarding emotions (i.e., sadness, disorientation, anxiety) and illness perception (e.g., perception of vulnerability, perception of control over MS) and 3 open questions regarding resources during the COVID-19 emergency. An extensive description of the survey is provided in a previous paper, including the full questionnaire as supplementary material [14].

The current study uses a new qualitative dataset regarding the last topic of the survey. Namely, “resources” in terms of benefit found were explored by asking participants to complete the following statement: “The most precious thing that the pandemic has taught me is”.

### 2.3. Procedure

The survey was made available through Limesurvey, an online tool that allows to collect anonymous responses. The invitation to take part in the survey was disseminated through social media (i.e., Facebook and Instagram pages and groups) dedicated to MS.

Data were collected between 13 May and 3 June 2020, immediately after the end of the country-wide lockdown imposed from 10 March to 3 May 2020. At the time of data collection, the Italian Government was gradually lifting the restriction measures. For example, starting from 4 May, only some industries re-opened and only visits to close relatives were allowed. Physical activities were only permitted open air, and funerals could only be held in small groups. Infection control measures, such as social distancing and the use of masks were compulsory, and people were advised to keep to a minimum social activities. From 18 May non-essential activities were re-opened, and people had more freedom of movement within regions, but only health- and work-related travel was allowed between regions.

### 2.4. Data Analysis

Descriptive statistics were presented as mean values and standard deviation (SD) for continuous variables and as frequencies for categorical variables. Student’s two-sample t-tests and Pearson chi-squared test were used to explore the differences between yawMS who reported at least one benefit finding and those who reported none.

As regards the qualitative analyses, participants’ answers were grouped in an Excel file and later analyzed using inductive thematic analysis [36]. Two raters (SP, VD) independently read all quotes and created an initial list of codes. These codes were then compared in a plenary meeting and concordant and discordant labels discussed with a third rater (MR). The codes were gradually categorized into initial themes. In the next stage, themes and sub-themes were examined, refined, and checked against the original data set, until consensus was reached, and all three raters were satisfied with the thematic map. As a next step, all answers were coded applying the finalized labels. Answers containing more than one benefit were divided into different quotes and coded separately. Frequencies of each theme were calculated. The interrater reliability revealed good agreement (i.e., percentage of agreement: 90% (CI 85.7–93.6%), Krippendorff’s Alpha: 0.88 (CI 0.84–0.93). After the coding, excerpts from the answers were then chosen to illustrate each theme and translated into English. The translation was performed by an author fluent in English (IMB) and checked for consistency with the other authors (MR, SP, VD). In a second phase, all three raters (SP, VD, MR) elaborated the themes into advice making them accessible as tips and suggestions for MS patients. Further validation and confirmation were obtained from a panel of patients (N = 13) who expressed, during two focus groups, their opinion on clarity, usefulness, and on how the tips reflected their experience.

According to the inclusion criteria of the ESPRIMO project, the definition of the age range for “young adults” from 18 to 45 years has been made on the basis of the clinical onset of MS (i.e., 20–40 years) and the course of the disease. Thus, our study focuses on a subgroup of MS patients who could be considered “young” given the disease history [36,37]. Regarding the second aim of the current research, we divided the sample into two sub-groups (i.e., ages 18–30, ages 31–45) and compared the frequencies of the theme and sub-themes between the groups using Pearson chi-squared test for independence of contingency tables, and the permutation approach was performed to adjust for cells with rare frequencies, where appropriate.

### 2.5. Ethical Considerations

As part of the “ESPRIMO” project [36], the present study was approved by the Ethical Committee of the Verona Hospital (Prog. 2676CESC) and registered on ClinicalTrials (ClinicalTrials.govID: NCT04431323). Informed consent was obtained from all subjects involved in the study. All responses were anonymous.

## 3. Results

### 3.1. Sociodemographic and Clinical Characteristics of the Sample

Of the 247 eligible respondents of the survey, 209 answered the question regarding benefit during the pandemic. Ten participants responded with “no learnings” or similar sentences and were therefore excluded from the qualitative analysis.

Respondents who provided at least one benefit (N = 199) had a mean age of 31.9 years (SD = 6.97) covering all possible ages (i.e., 18–45 years); 46% of respondents were 18–30 years old. Of all respondents, 95 (45%) were married or living with a partner. The majority (58%) was employed, 14% were students, and the remaining respondents (30%) were not occupied (i.e., jobless, searching for a job, occasional work or retired) or did not answer the question. Regarding the place of residence during the COVID-19 pandemic, respondents were from all Italian regions, with almost half of yawMS (46%) living in the northern, 20% in the central and 35% in the southern regions. Regarding gender, unfortunately, due to a technical problem with the survey only 22% of data were available, as indicated in the table in Appendix A. Most participants reported a diagnosis of relapsing-remitting MS (187; 94%), whereas 4 (2%) had primary progressive MS, 5 (3%) secondary progressive MS and 3 (2%) Clinically Isolated Syndrome (CIS).

As regards participants’ emotions during the pandemic, respondents rated their anxiety/worry about the course of their disease, sadness/discouragement linked to MS, and disorientation/confusion about managing the disease on a scale of 1 to 10 with a mean value of 6.5 (SD = 2.4), 6.1(SD = 2.7), and 5.8 (SD = 2.8), respectively. Regarding the impact of the pandemic on patients’ illness perception, respondents rated their feeling of vulnerability on a scale of 1 to 10 with a mean value of 6.9 (SD = 2.4) and their feeling of personal control with a mean value of 5.6 (SD = 2.3).

Inferential statistics failed to highlight differences in sociodemographic and in participants’ emotions during the pandemic between participants who provided at least one benefit (N = 199) and the others (N = 48).

### 3.2. Qualitative Results

The total number of quotes including at least one benefit was 220, with 20 respondents reporting more than one.

Qualitative content analysis identified three distinct themes that represented benefit finding associated with the COVID-19 pandemic experience: (1) personal growth (2) relational growth, (3) existential growth. The first and the third theme are further divided into subthemes. Themes and subthemes, their frequencies, and exemplary quotes are listed in Table 1.

Existential growth was the most frequent theme, reported in 42% of the quotes, followed by personal growth, reported in 33% of the quotes, and by relational growth, reported in 25% of the quotes. Regarding the subthemes of existential and personal growth, most of the quotes regarded valuing “small things”, slowing down/mindful attitude, adopting adaptive coping, and taking care of oneself.

In the following part, each theme is presented with the respective subthemes and exemplary quotes, translated from Italian. Each quote is coded to denote the participant’s ID and age (i.e., ID-age).

#### 3.2.1. Theme 1: Personal Growth

This theme includes five subthemes: (i) Taking care of oneself; (ii) Adopting adaptive coping; (iii) Becoming aware of one’s personal strengths; (iv) Acknowledging solitude and independence; (v) Recognizing the value of what you have.

Taking care of oneself. Learning benefits in the personal area regarded the importance of taking care of (“*That I have to love myself, always*”, 9—age 35) and taking time for oneself. In this context, practical activities such as cooking, exercising, or doing other hobbies (“*That I have to take more time for myself and for pursuing my hobbies*”, 49—age 38) but also more intimate aspects were mentioned. As for the latter, some participants recognized the importance of listening to personal needs and rhythms, taking time for oneself and one’s own interests, as exemplified by the following quotes: “*Respect my needs*” (73—age 34); “*The most valuable thing this quarantine has taught me is that I can give myself time*” (385—age 29); “*The time we take for ourselves is never enough; we should learn not to rush all the time and focus on what we really need-ourselves; taking a break can only help you be more productive*” (276—age 30).

Adopting adaptive coping. Several participants stated that they had learned strategies that could be employed not only during the pandemic but also to face illness or other stressors in their daily life. For example, as stated by one participant, one learning during the pandemic was that “*You must first accept that you face difficulties in order to really solve them*” (181—age 39). Three quotes regarded specifically the importance of starting to accept MS. As one participant said (365—age 39): “*That I have to accept the disease because if I fight against it, I won’t solve anything. It is now part of me*”.

Among the adaptive strategies, flexibility and openness to change were mentioned “*The ability to adapt. Knowing how to reinvent myself for my mental/physical health*” (272—age 24) as well as self-esteem (“*Believe in yourself*”, 291—age 27) and prioritizing personal goals and “*Thinking about what you really want*” (138—age 27).

Emotional regulation expressed as the ability to listen, to recognize and acknowledge emotions and feelings (“*Understanding when I am stressed*”, 120—age 34) together with the commitment to leave behind negative emotions (“*that anxiety is useless*”, 220—age 38) was also listed among the adaptive strategies. Other examples of strategies are: “*We need to talk less and think more*” (141—age 26); “*That we cannot plan anything lightly. We have to be objective and realistic facing day after day*” (20—age 32).

Becoming aware of one’s personal strengths. Several benefits focused on becoming more aware of personal strengths. YawMS rediscovered and reinforced the awareness of their inner strength in the face of the pandemic, as described in the following example “*I’m stronger than I thought*” (366—age 35). Participants realized that they are able to cope with difficulties despite certain weaknesses and to adapt to the changes imposed by the pandemic, as illustrated by this quote: “*it made me realize my great spirit of adaptation*” (362—age 28). The pandemic has also taught participants the value of empowerment when dealing with changes, as noted by one individual “*I can decide which turn to give to my days. I can be happy and productive even when being at home*” (347—age 29).

Some yawMS also described becoming aware of their inner strength in relation to MS, (“*You are the one being in charge of your life, not MS, it is only an accessory*”, 149—age 22) and in relation to personal achievement: (“*It was knowing and understanding the new me. Thanks to the quarantine I stopped and I felt certain that I had changed for the better*”, 24—age 33). Linked to this renewed awareness of personal competences, one participant (176—age 40) said that “*despite everything, I deserve to be happy*”. 

Acknowledging solitude and independence. Being alone and independence represented another subtheme. Social distancing measures and mobility restrictions imposed by the Italian Government to contain the spread of the virus forced some people to live alone with limited opportunity for social encounters. Despite these challenges, yawMS reported that during the pandemic they have managed to learn to be alone with themselves. Indeed, some participants positively evaluated this experience of isolation, which gave them the opportunity to learn more about themselves, and re-framed it as an opportunity of personal growth, as illustrated by these examples: “*Learn to live with myself, with silence and solitude*” (122—age 34); “*Being alone is good, you get to know yourself better*” (169—age 22). Some participants considered being alone also as a possibility to be independent: “*Having found physical and mental serenity in solitude and the thought of not having to depend on anyone*” (97—age 30).

Recognizing the value of what you have. Finally, participants reported a greater appreciation of what they have as a consequence of the pandemic experience both regarding their jobs and their home, acknowledging the positive effect of working and living in a satisfying place, as illustrated by the following statements: “*That being able to work is important to occupy the mind and decrease negative thoughts*” (21—age 41); “*I appreciate very much the place where I live (countryside near the city)*” (304—age 22).

#### 3.2.2. Theme 2: Relational Growth

About one quarter of quotes regarded interpersonal relationships. Participants stated to have become more aware of the importance of close relations and connection with people.

Specifically, participants perceived a sense of gratitude towards parents, partners, friends, and children, reporting a strengthening of closer relationships. Relationships with family members were recognized as particularly relevant.

The importance of having a person around who is present, able to listen and to take care has been highlighted: “*The importance of affection and understanding of those around you*” (46—age 41); “*Being surrounded by positive people who have coped with this situation without being overwhelmed by fear and despair*” (119—age 27). Together with this benefit, participants said that they became more aware of the real nature of their relationships, since critical situations can enable genuine expression of loving and supporting behaviours, as illustrated by the following quotes: “*That the people who love you are always there*” (150—age 32); “*Who really supports and helps me*” (10—age 25).

Moreover, the value of significant relations should not be taken for granted. Acknowledging this is essential, in the view of yawMS, especially in difficult situations, as one participant put it: “*Rediscovery of previously obvious relationships*” (34—age 28).

The relevance of the connection with others regarded also physical contact as demonstrated by the following quotes: “*The value of a hug, the importance of someone’s presence, especially in certain moments or circumstances*” (22—age 45); “*In recent months, there have been meetings, dinners, afternoons, even new friends, via Skype. I have true, healthy friendships who understand and support me. Isolation is not a limit, although nothing beats the beauty of a hug*” (107—age 36).

Finally, one participant emphasized the importance of taking care of significant others: “*The importance of taking care of loved ones*” (13—age 33).

### 3.3. Theme 3: Existential Growth

The most frequent category reported by almost half of the participants includes general benefits about life and human values. In this category, 6 subthemes were identified: (i) Valuing “small things”; (ii) Slowing down/mindful attitude; (iii) Acknowledging human vulnerability; (iv) Discovering freedom; (v) Valuing health; (vi) Other learnings about life and human values.

Valuing “small things. Several yawSM rediscovered during the pandemic the value of small things that are sometimes considered of little worth. As one participant put it (130—age 33), “*even small, everyday things have a priceless value*”. The restrictions and the rules imposed by the health emergency changed drastically the life of Italians, especially in the aftermath of the pandemic when the survey was conducted. As a twenty-year-old participant said, it is the lack of something that helps to take a different perspective: “*That small things, when they are not there, become huge. And therefore, they must be appreciated, and must be enjoyed to the fullest, when possible*” (199—age 22). As a result, nothing should be taken for granted and even what is considered normal deserves to be appreciated, as illustrated in these quotes: “*The value of what we usually consider normal*” (246—age 43); “*Never take anything for granted*” (298—age 22).

Slowing down/mindful attitude. Participants stated in multiple quotes that they have learned to appreciate time and to slow down. Participants recognized “*How much precious time we waste*” (289—age 39) and “*That lost time does not come back*” (343—age 28), every single moment is therefore extremely valuable and so it should not be wasted. Participants pointed out the importance of making the best out of every moment, of seizing the day, and of living in the present: “*Enjoy every moment of life*” (152—age 39); “*Living day by day, without plans, deadlines, expectations*” (60—age 26).

The value of patience (“*Learning or trying to be patient*”, 343—age 28), and the importance of experiencing time with a calm and mindful attitude are also reported and included in this subtheme category: “*The beauty of tranquillity*” (50—age 30).

Acknowledging human vulnerability. In the aftermath of the pandemic, participants recognized the vulnerability of human beings and the frailty of life, as illustrated by the following examples: “*We are all vulnerable, we do not control our lives*” (292—age 18); “*Life goes by in a flash”* (109—age 38). Linked to this vulnerability, participants also reported a sense of uncertainty, as expressed by this participant: “*I have learned that life is unpredictable*” (370—age 29).

Discovering freedom. One of the other subthemes regards “*The importance of freedom*” (197—age 32). Freedom was seen as highly relevant also in relation to the disease, as expressed in this quote: “*That the freedom and the possibility to choose, to do, to manage one’s daily life is important and fundamental for feeling good in general, but above all for accepting the disease and facing the difficulties that derive from it*” (250—age 26).

Valuing health. Some participants reported that during the pandemic they have become more aware about the relevance of health, as an important aspect of life that should be preserved, as illustrated by the following statement: “*Protecting your health is always the most important thing*” (88—age 42).

Other learnings about life and human values. Other themes were reported by a minority of participants regarding various aspects of life and human values, for example, the love for nature (“*the love for nature*”, 317—age 41), the relevance of research (“*that research, in any field, is vital*”, 140—age 43), the importance of values, such as responsibility and respect (“*Respect for others*”, 66—age 34), as well as the value of equality linked to a common emergency situation (“*That we are all equal*”, 190—age 45) and the power of being connected (“*That union is strength*”, 195—age 43).

### 3.4. Comparison of Participants’ Characteristics and Qualitative Results between Age Groups

The analysis by subgroups have been performed on the respondents who provided at least one benefit (N = 199).

Regarding the sociodemographic characteristics (see full table in Appendix A), no differences between the two age groups were found in terms of educational status (X^2^_(1)_ = 0.32 *p* = 0.57) and area of residence (X^2^_(2)_ = 1.36 *p* = 0.51). The participants in the younger group (aged 18–30 years) were more frequently employed (X^2^_(1)_ = 14.26 *p* < 0.01), and less frequently married (X^2^_(1)_ = 33.21 *p* < 0.01) than those in the older group (aged 31-45 years). A slight significance emerged in the distribution of gender (X^2^_(1)_ = 4.38 *p* = 0.04), with more men belonging to the younger group. However, as already stated, gender was explored only in a sub-sample of participants. No significant differences between the age subgroups were found in terms of diagnosis (X^2^_(3)_ = 6.23 *p* = 0.1). T-tests did not reveal any differences between these groups with respect to anxiety, sadness, disorientation, feeling of vulnerability, and feeling of control (see table in Appendix A). 

Similar percentages of the quotes were found for the two age groups (18–30 years of age 46% and 31–45 years of age 54%). Figure 1 shows the frequencies of the quotes for each of the subthemes of existential and personal growth, and for relational growth for the very young adults under 30 years and those between the ages of 31 and 45. No significant differences between age subgroups were found at the theme level of coding (X^2^_(2)_ = 1.06 *p* = 0.59) and, within the existential and personal growth themes, at subthemes level: Personal Growth (X^2^_(4)_ = 0.92 *p* = 0.94) and Existential Growth (X^2^_(5)_ = 4.06 *p* = 0.55).

## 4. Discussion

The present cross-sectional study aimed to explore benefit finding among young adults living with MS during the COVID-19 pandemic. Despite the negative psychological impact experienced by yawMS in terms of emotions and perceptions regarding MS (see Donisi et al. [14] for a full description of psychological distress linked to the pandemic in our population), the majority of the participants in the survey reported at least one benefit finding during the pandemic. Benefits reported by yawMS in the face of COVID-19 regarded different aspects of the self, relationships and general values, including both concrete and abstract learnings. Applying qualitative content analysis, benefits were categorized into three main themes: personal growth, relational growth and existential growth. Existential growth was the most frequent theme. It includes a change in the perspective on values, such as freedom and health, an increased awareness of the importance of time, and the recognition of human vulnerability. The second most frequent benefit was personal growth, encompassing different areas of the self, such as increased awareness of personal and material resources and of the importance of finding strategies to adapt and improve personal wellbeing. The least frequent theme, found in a quarter of the responses, regarded relationships and the importance of human connection and closeness with other people.

As regards our second aim, namely quantitative comparisons between the frequencies of quotes, the findings showed that younger MS patients (18–30 years old) do not differ from those between 31 and 45 years of age regarding the identification of benefits when facing adversities. Despite the relatively wide age range and the potential differences in terms of psychosocial needs and roles between the two age groups, it seems that the reaction to the pandemic among young adults with MS was quite homogeneous. Consistently, the two age groups did not significantly differ with respect to illness perception and emotional response, as already suggested by a previous paper on the same population that outlined similar levels of distress [14].

Regarding the benefit finding related to age, interestingly, a meta-analysis demonstrated that younger age is associated with more benefit finding [38]. Future studies should be extended to other age groups in order to explore benefit finding also in older populations. In fact, older adults may have particular experiences with the pandemic and quarantine (e.g., higher risk of complications from COVID-19, greater isolation) which could determine quantitative and qualitative differences in terms of benefit finding.

The psychological impact of COVID-19 in terms of stress, negative emotional reactions (e.g., anxiety, depression), and traumatic responses has been extensively studied in the general population [39] and to a lower extent also in vulnerable populations, such as chronically ill patients, including patients with MS [15,17,18,19,20,21,22,23]. Moreover, young adults are particularly vulnerable and at greater risk of developing adverse psychological symptoms during the pandemic, as demonstrated in a study by Forte and colleagues [40] that focused on people under the age of 50. Similarly, a recent review found higher rates of depression, anxiety, and distress in individuals under 40 years of age [6]. However, only a few studies investigated potential positive outcomes of living through a potentially traumatizing time as the current one. To the best of our knowledge, only one recent study investigated benefit finding during the COVID-19 pandemic, highlighting, in line with our results, that college students found a wide range of benefits [41]. The psychological construct of benefit finding has been explored in a variety of negative and potentially traumatic situations, such as war, terrorism, natural disaster, bereavement, and chronic illness [38].

Considering personal growth subthemes, increased awareness of personal strengths, improvement in adaptive coping and opportunities to reprioritize certain aspects of life emerged as common themes in our study, as in the study by August and Dapkewicz [41] and are also common benefits perceived after other negative events [42]. Similarly, appreciating what you have has been mentioned both in relation to COVID-19 [41] and in response to having MS [43].

The positive evaluation of being alone as an opportunity to learn more about oneself, which has been identified both in our study and in the study on college students [41], might be a peculiar benefit of the current pandemic. Indeed, it has been argued that benefit finding is context-specific since the areas of benefit may vary with respect to the type of adversity [43] and social distancing and isolation are, in fact, distinctive characteristics of the COVID-19 pandemic. As regards this subtheme of personal growth, it seems possible to feel solitude (being at peace with oneself) rather than loneliness, a negative experience of fear and isolation which represents a risk factor for mental illness [44]. In Italian, there is only one term to refer to loneliness which usually has a negative meaning. However, our participants referred to the experience of being alone and to loneliness as an opportunity to learn more about themselves and grow from it, clearly reframing it in a positive way.

Further, the importance of taking care of themselves was a benefit reported by many participants. Although this area has not been broadly described in other studies, there are some similarities between our study and a qualitative study on MS patients by Pakenham [32], which found an increased desire to listen to one’s own needs as a consequence of living with the disease.

Relational growth has been reported in previous qualitative and quantitative studies exploring benefit finding [32,42,45]. The increased awareness of the importance of relationships is particularly for the COVID-19 pandemic as containment measures have reduced the opportunities of socialization, in some cases increasing social isolation. On the other hand, especially in the first wave of COVID-19 in Italy, when strict stay-at-home measures were implemented, some families spent more time together. Since people have experienced relationships in different ways than usual and to a different extent, they may have adopted new perspectives and become more aware and appreciative. Indeed, participants of the present study perceived a sense of gratitude towards significant others and highlighted the importance of nurturing personal relationships, which can have a positive impact on mental health. In fact, perceived social support has been linked to greater resilience to stress [46], an association recently confirmed in the general population during the pandemic [47]. Although the present study did not aim to verify the association between social support and distress, it could be hypothesized that finding ways to experience closeness and emotional connection with others, despite severe social-distancing measures, might be a crucial element to adapt to negative life events. This might be particularly relevant in patients with MS since social support has been linked to better adjustment outcomes to the disease [48].

In contrast to other studies focusing on MS [43] or on general life stress [45], participants in the present study did not report an increased sense of compassion or empathy towards others. The lack of explicit reference to empathy and giving might be due to the measurement used in the current study. Indeed, the present study used open-ended questions instead of a list of possible closed-ended answered typically used in questionnaires, which are often prone to social desirability bias as reported by Pakenham and Cox [43].

In the area of existential growth, one of the most frequent learnings regarded the greater significance attributed to small aspects of everyday life. The conditions impacting the quality of life and the consequent drastic changes in routines might have helped participants to fully appreciate the value of what is usually underestimated. Some participants highlighted the importance of health, a theme that might be particularly important for patients with MS and chronic illness [43] since it did not emerge in the above-mentioned study on healthy college students [41]. As already underlined, benefit finding is context-specific. However, our results seem to suggest that it may vary also depending on personal characteristics. In fact, it could be argued that people with a chronic illness are more inclined to think about health, also due to a possible perceived vulnerability to COVID-19. Consistently, some participants recognized the frailty and vulnerability of human beings, a theme that can be seen as connected to the importance of health. It must be noted that in our study benefits in the health area referred to human beings in general and were not related to the self, as reported in other studies on benefit finding linked to MS diagnosis [32,43]. Although the pandemic was omnipresent as a widespread health crisis impacting the health of people worldwide and our question specifically asked about it, it is interesting to note that few participants directly mentioned MS in their answers.

Another benefit finding that could be specific to patients with MS and patients with chronic illness is a mindful attitude as it has been found both in the present study regarding the experience of COVID-19 in MS patients and the experience of illness in MS patients [43].

As stated in the introduction, the clinical purpose of this qualitative exploration was to develop patient-generated tips based on meaning-making and benefit finding to help other young adults with MS to face the pandemic and, more in general, their illness and its impact on their quality of life. Indeed, the present study is part of the ESPRIMO project that aims to improve the quality of life of yawMS through the collaboration between patients living with the disease, clinicians, and researchers [36]. Figure 2 presents patients’ tips by drawing on the metaphor underlying the ESPRIMO logo, namely a sailing boat navigating in a storm, that symbolically values the resilience potential of vulnerable populations (i.e., if you are in the middle of a storm, you cannot control the wind, and you are navigating a fragile boat, all you can do is adjust the sails and use the wind in your favour). Visual methodologies are a novel and fruitful approach to qualitative research in medical settings with different purposes [49]. In particular, the use of visual metaphors leads to greater involvement and engagement, as the message is conveyed in an emotionally salient, symbolic, and inspirational way. Their successful use in previous campaigns on mental health suggests that they are an effective strategy for communicating complex health messages [50].

To disseminate the tips to patients and clinicians. For example, the picture will also be distributed to clinicians of the MS regional clinical centre of Verona (involved in the ESPRIMO project) to be used in the clinical encounters with their MS patients. Focusing on positive outcomes is important in clinical practice since patients should be reminded that being vulnerable does not mean being hopeless and that even in the face of a long and disruptive storm, people can take advantage of the wave, regardless of the strength of their boat. Clinicians, according to bio-psychosocial medicine [51], should be aware of potential negative outcomes of chronic illness on psychological wellbeing but should also consider that adverse life events may foster meaning-making. Being able to recognize and validate cues of positive processes of adaptation during the clinical encounter could be beneficial for fostering patients’ wellbeing starting from their personal perspectives and values. Clinicians should encourage patients to take an active role in the management of the disease, focusing on autonomy, self-efficacy, and specific skills in managing their health [52]. Supporting patients in finding their own ways of mobilizing their strengths and resources is essential in providing patient-centred and empowering care for people with chronic illness [53].

Finally, the results of the presents study will inform the development of the specific bio-psycho-social co-created intervention for yawMS as part of the ESPRIMO project. Involving patients and considering their experiences and perspectives is crucial for planning and implementing tailored interventions. Benefit finding has already been part of an intervention aimed at improving resilience of young patients with cancer [54] and has been found to reduce depressive symptoms in caregivers of patients with Alzheimer’s disease [55]. Thus, including benefit finding in psychological interventions for patients with MS could be beneficial also for this particular group of patients.

The current research is particularly innovative as it explored the positive effects, such as benefit finding, in the aftermath of the pandemic. Moreover, the present study focused on the experiences of young adults with MS, who are particularly vulnerable in terms of adjustment to the disease and highlighted that people might experience benefits in the face of a difficult event regardless of their psychological vulnerability. Although few recent papers have discussed the psychological impact of patients with MS or other chronic diseases, to the best of our knowledge no other study, apart from our previous paper [14], has focused on a group of people having potential double vulnerabilities (i.e., being young with a chronic disease).

The present study was conducted in a specific time frame immediately after the peak of the pandemic. It must be considered that the literature suggests that time since the experience of a trauma is not related to benefit finding but that perceived stress is associated with more benefit finding [38]. Therefore, participants’ answers regarding benefit finding might have been different if collected in another period of the pandemic, for instance, during the lockdown which was probably associated with more stress and anxiety. However, it has to be highlighted that questions were retrospective, as they focused on what participants have learned during those hard times.

The present study is cross-sectional and did not aim to evaluate the stability of benefit finding and its impact on wellbeing and adaptation over time. Future studies using a longitudinal design are needed to assess the potential qualitative and quantitative changes in the perception of benefit finding during the traumatic event and after months or years, also measuring actual changes in behaviours.

The use of social media for the dissemination of the survey has strengths and limitations. Indeed, it is known to be a very popular channel of interaction for young adults [56] and a powerful tool to recruit participants and thus helped us to reach a large pool of potential participants across Italy. However, using social media as a recruitment method is often linked to selection bias, which can reduce the generalizability of the findings. Indeed, patients who did not use social media or are not active in online MS groups were not able to respond to the survey. Moreover, patients who were not interested in psychological topics were probably less likely to participate.

It must be also considered that almost all respondents reported a diagnosis of relapsing–remitting MS, and patients with other types of MS were less represented. Our focus on young people with MS has excluded patients older than 45 years from our study population, leading to a reduced representation of cases with secondary progressive and primary progressive MS, which are known to begin at an older age than relapsing-remitting MS. This likely led to the exclusion from our sample of more disabled MS patients on average, who might have different needs, psychological reactions, and adaptive strategies in the context of COVID-19 pandemic in comparison with less disabled patients. Therefore, the results of this survey should not be extrapolated to patients with progressive forms of MS. The generalizability of the findings could be further limited by the lack of control on the representativeness of the sociodemographic characteristics of the study sample in respect to the broader MS Italian population. As already stated in our previous paper [14], another shortcoming of the present study is the low percentage of answers to the question regarding gender.

Further, even if the use of an online survey was in line with the procedures of infection control and was appropriate considering the explorative nature of the study, it did not allow controlling the fulfilment of the inclusion criteria that were self-evaluated by the participants and potentially limited the length of answers for the qualitative answers. However, the open qualitative exploration of patients’ perspective on benefits finding enabled us to capture a wide and genuine description of all learnings derived from the pandemic, reducing the risk of social desirability response bias. In fact, patients answering to precompiled questionnaires, such as the Perceived Benefit Scale [42], could be more prone to select more items related to benefit finding than the ones they thought of. On the contrary, the open question format gives respondents more freedom, also facilitating the reports of no benefit.

## 5. Conclusions

Psychological growth can be a possible outcome when living through a pandemic. In fact, the majority of the participants in the survey reported at least one benefit finding regardless of the perceived psychological impact of the pandemic. Benefit regarded growth in the personal, relational, and existential area. The identification of benefits did not differ between very young adults under 30 years and those between the ages of 31 and 45.

The present study highlights that positive psychological adaptation in the face of adversity can also occur in vulnerable populations, such as young adults with MS. Psychological support programs should focus not only on reducing the risk of developing maladaptive responses but also on promoting meaning-making, for instance, framing difficulties as an opportunity to find benefit. Using an approach that focuses on resources and adaptation to the disease and adversity can foster resilience processes and empower patients.

## Figures and Tables

**Figure 1 ijerph-18-04156-f001:**
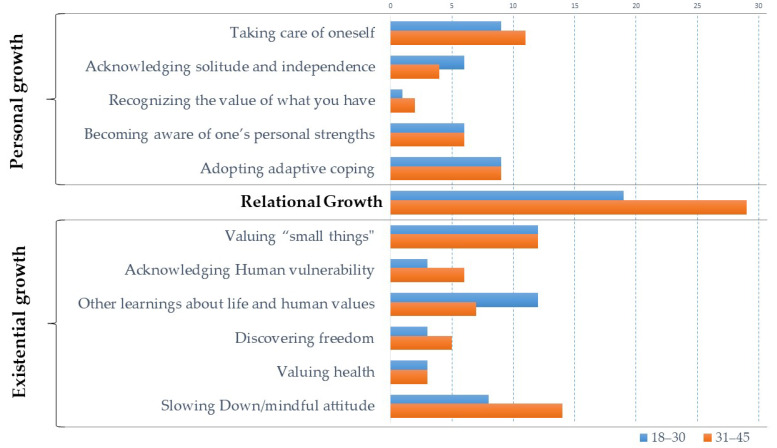
Frequencies of quotes by the two age subgroups (i.e., 18–30; 31–45).

**Figure 2 ijerph-18-04156-f002:**
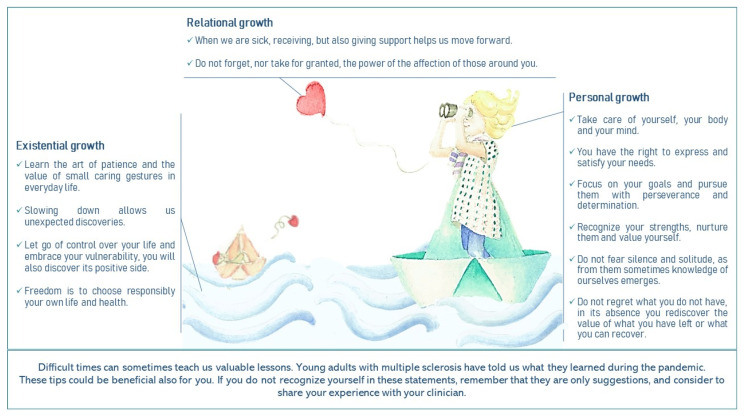
Results on benefit finding translated into tips for patients.

**Table 1 ijerph-18-04156-t001:** Frequencies of themes and subthemes.

Themes	Subthemes	Exemplary Quotes	Number of Quotes (%)
Personal growth	Taking care of oneself	*“Dedicate more time to myself”*	21 (10)
	Adopting adaptive coping	*“* *You must first accept that you face difficulties in order to really solve them* *.”*	22 (10)
	Becoming aware of one’s personal strengths	*“* *It made me realize my great spirit of adaptation”*	13 (6)
	Acknowledging solitude and independence	*“Being alone is good, you get to know yourself better”*	11 (5)
	Recognizing the value of what you have	*“Always be serene and happy with what you have.”*	5 (2)
Relational rowth		*“The importance of affection and understanding of those around you”*	55 (25)
Existential growth	Valuing “small things”	*“That small things, when they are not there, become huge. And therefore, they must be appreciated, and must be enjoyed to the fullest, when possible* *”*	25 (11)
	Slowing down/mindful attitude	*“Enjoy every moment of life”* *“The beauty of tranquillity”*	23 (11)
	Acknowledging human vulnerability	*“We are fragile as human beings”*	9 (4)
	Discovering freedom	*“Appreciate the gift of freedom”*	9 (4)
	Valuing health	*“Protecting your health is always the most important thing”*	7 (3)
	Other learnings about life and human values.	*“The beauty of the world without pollution”* *“Respect for others”*	20 (9)
		Total	220 (100%)

## Data Availability

The dataset presented in this article is available from the corresponding author on reasonable request.

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
