# Peer review of "“If You Can’t Control the Wind, Adjust Your Sail”: Tips for Post-Pandemic Benefit Finding from Young Adults Living with Multiple Sclerosis. A Qualitative Study"

_ijerph, 2021, doi:10.3390/ijerph18084156_

Round 1
Reviewer 1 Report
In the material and methods section the data is mixed and it is not clear.
This section should include three clear sections:
- participants (population to which the survey was sent)
- instrument used to collect data
expose the study variables and bibliography.
3.procedure (insert the link of the survey, time in which it was passed and situation of national confinement measures that were used during the time)
- Data analysis
as it is
- Ethical considerations
Remember to write ethical considerations at the end of the material and methods
Conclusions
Must be redone, they do not respond to silver targets on line 91-96
Include the following articles in introduction or discussion
Reguera-García, M. M., Liébana-Presa, C., Álvarez-Barrio, L., Alves Gomes, L., & Fernández-Martínez, E. (2020). Physical activity, resilience, sense of coherence and coping in people with multiple sclerosis in the situation derived from COVID-19. International journal of environmental research and public health, 17(21), 8202.
Umucu, E., & Lee, B. (2020). Examining the impact of COVID-19 on stress and coping strategies in individuals with disabilities and chronic conditions. Rehabilitation psychology.
Akhoundi, F. H., Sahraian, M. A., & Moghadasi, A. N. (2020). Neuropsychiatric and cognitive effects of the COVID-19 outbreak on multiple sclerosis patients. Multiple sclerosis and related disorders.
Moghadasi, A. N. (2020). Evaluation of the level of anxiety among Iranian multiple sclerosis fellowships during the outbreak of COVID-19. Archives of Iranian medicine, 23(4), 283.
Reviewer 2 Report
This paper describes research into a very worthwhile topic, focusing on the experience of benefit finding in young people with MS.
I have a few minor comments and suggestions.
Introduction
This presents a good overview of background literature and sets up the rationale for the study well.
One general query would be why the authors split participants up into 18-30 vs 31-45, as often population categories are split into 18-24, 25-34 etc. Is there a reason in the context of MS as to whether there may be a different set of experiences over 30?
The introduction could also benefit from some general broad context on MS (e.g. types of MS such as RRMS, SPMS, PPMS etc.). Given that proportion of secondary progressive MS increases with age (including increasing disability), this is perhaps something that might be considered in the context of this study?
Method
As this was a convenience sample, did the authors check whether sample characteristics were representative of the broader MS population? If not, it would be worth including this as a potential limitation in the discussion section
Given that the study was collected in Italy (and presumably all responses were in Italian), could something be said about the process of translating quotes and whether this was done prior to or after coding?
Results
Demographics are described for group as a whole, but were there any differences here between the two age categories? Similarly, were there any differences on perceptions of vulnerability or personal control between the two age groups?
I also didn’t see mention of the gender breakdown of this sample, which would be important to mention and discuss in the context of benefit finding.
There is a nice distinction drawn between personal growth, relational growth and existential growth, with some lovely examples of quotes provided from participants.
Discussion
The findings are discussed in depth, with good reference to appropriate literature.
The inclusion of “tips” for patients based on responses is also a very nice addition. I wonder prior to sharing this picture, might it be worth getting further feedback from a patient panel as to whether this reflects their experience/sentiments? This may be something that is already built into your project, but if not it might be something to consider.
Although it is clear that the focus of this study is on younger adults with MS, it might also be worth considering whether similar findings might be expected for older adults living with MS? Given that they may be at higher risk of complications from COVID-19, would the authors expect that benefit finding would take a similar form in this group?
Also, it is noteworthy that data collection took place in the summer of 2020, when COVID-19 restrictions may have been beginning to ease, with case numbers falling compared to March/April 2020 (at least temporarily). Do the authors think that responses may have been different if they had been collected in a more restrictive period of lockdown?
Aside from these minor points, I would like to thank the authors for writing this manuscript – I really enjoyed reading it!
Round 2
Reviewer 1 Report
all requested changes have been made or justified. You have my support for your publication, good work and a lot of strength to continue working on this issue.
Author Response
We would like to thank the reviewer for his/her comments.